# Control Simulation Experiment with the Lorenz's Butterfly Attractor

Takemasa Miyoshi[1,2,3,4], Qiwen Sun[1,5]

[1]RIKEN Center for Computational Science, Kobe, 650-0047, Japan
[2]RIKEN Cluster for Pioneering Research, Kobe, 650-0047, Japan
[3]RIKEN interdisciplinary Theoretical and Mathematical Sciences (iTHEMS), Wako, 351-0198, Japan
[4]Application Laboratory, Japan Agency for Marine-Earth Science and Technology (JAMSTEC), Yokohama, 236-0001, Japan
[5]Graduate School of Mathematics, Nagoya University, Nagoya, 464-8601, Japan

*Correspondence to*: Takemasa Miyoshi (takemasa.miyoshi@riken.jp)

**Abstract.** In numerical weather prediction (NWP), the sensitivity to initial conditions brings chaotic behaviors and an intrinsic limit to predictability, but it also implies an effective control in which a small control signal grows rapidly to make a substantial difference. The Observing Systems Simulation Experiment (OSSE) is a well-known approach to study predictability, where "the nature" is synthesized by an independent NWP model run. In this study, we extend the OSSE and
design the control simulation experiment (CSE) where we apply a small signal to control "the nature". Idealized experiments with the Lorenz-63 three-variable system show that we can control "the nature" to stay in a chosen regime without shifting to the other, i.e., in a chosen wing of the Lorenz's butterfly attractor, by adding small perturbations to "the nature". Using longer-lead-time forecasts, we achieve more effective control with a perturbation size less than only 3% of the observation error. We anticipate our idealized CSE to be a starting point for realistic CSE using the real-world NWP systems, toward
possible future applications to reduce weather disaster risks. The CSE may be applied to other chaotic systems beyond NWP.

## 1 Introduction

The "butterfly effect", discovered by Lorenz in 1960s (Lorenz 1963; 1993), is a phenomenon that an infinitesimal perturbation like "a butterfly flapping its wings in Brazil" causes a big consequence like "a tornado in Texas". This extreme sensitivity brings chaotic behaviors and an intrinsic limit to predictability, but it also allows to design an effective control
which was explored as "the control of chaos" in 1990s (e.g., a review by Boccaletti et al., 2000). Namely, we could take advantage of "the butterfly effect" and design an effective control with a series of infinitesimal interventions leading to a desired future. The control of weather is human's long-time desire, and if we know when and where to put a "butterfly", we could lead a better life by, for example, reducing the risks of tornadoes.

Predictability has been studied extensively, and we enjoy current high-quality weather prediction being consistently
improved. However, studies on controllability are limited because we had to first improve the prediction accuracy and because our engineering power may be insufficient to enforce large enough perturbations to the atmosphere. Based on recent

high-quality NWP, this study attempts to explore a computational simulation approach to weather controllability. The simulation studies reveal what perturbations are needed to modify and control the weather. Mutual interactions between the simulation studies and the intervention techniques would be essential for the future developments toward real-world applications.

Previous efforts on weather modification include rain enhancement studies (e.g., a review by Flossmann et al. 2019) by cloud seeding with ground-based facilities and aircraft injecting smokes and dry ices into moist air, so that the aerosols act as cloud condensation nuclei and enhance cloud formation. These studies greatly helped advance our knowledge about physical processes of clouds and precipitation, but in terms of controlling the weather, we had only limited success with unclear implications to high-impact weather events, mainly because this method works only with supersaturated air. In the climate scale, geoengineering is a widely discussed concept, such as launching mirror satellites to reflect the sunlight and injecting dusts into the stratosphere to block the sunlight for cooling the air. Li et al. (2018) performed computational simulations and explored potential rain enhancements in the Sahel region by implementing large scale wind and solar farms over Sahara Desert and modulating the global atmospheric circulation. However, actual geoengineering operations are controversial because they may cause irreversible unexpected side-effects due to our limited knowledge of the earth system. The accepted and currently ongoing operations to counteract the current climate change may be limited to reduce the greenhouse gas emissions and to enhance renewables and recycles.

Our focus here is different. We aim to apply "the control of chaos" to the weather. We do not aim to cause a permanent irreversible change to the nature, but we would like to control the weather within its natural variability and to aid human activities, for example, by shifting the location of an extreme rain region to avoid disasters without causing a side effect to the global climate. For extreme weather that occurs in a chaotic manner under natural variations, the control of chaos suggests that proper infinitesimal perturbations to the nature atmosphere alter the orbit of the atmospheric dynamics to a desired direction. If the proper infinitesimal perturbations are within our engineering capability, we could apply the control in the real world. However, we cannot be too cautious about potential side effects and must consider and address every possible consequence. We will come back to this issue later in conclusion.

Here we develop a method of the control simulation experiment (CSE). It would be straightforward to extend the method to broader fields with chaotic dynamics beyond NWP. Weather prediction has been improved consistently by studying predictability and better initial conditions for NWP. Data assimilation (DA) combines the NWP model and observation data for optimal prediction. The method of DA shares that of optimal control, such as the Kalman filter (Kalman, 1960), where prediction and control are the two sides of a coin. DA has been studied extensively to improve the prediction, and this study illuminates the control.

The OSSE is a powerful method to simulate an NWP system (e.g., Atlas, 1985; Hoffmann and Atlas, 2016). The OSSE can be designed to assess the impact of certain observing systems and is useful, for example, to evaluate the potential value of a new satellite sensor before launch. The OSSE can also be designed to evaluate DA methods. In the OSSE, an independent model run acts as a synthetic "nature run" (NR), and we simulate observations by sampling the NR. The NWP

system is blind to the NR, takes the simulated observations, and estimates the NR. We compare the estimation accuracy among different OSSEs with different observations and different DA methods.

Here we extend the OSSE and apply small perturbations to the NR to alter the orbit to a desired direction. Investigating effective perturbations would address the controllability. As a proof of concept, we focus on the essence of the problem and use the Lorenz's three-variable model (L63, Lorenz, 1963) instead of using a complex large-scale NWP model. In predictability studies, OSSEs are often performed with such simple idealized models like L63 to explore new DA methods before applying to the real NWP models (e.g., Kalnay et al., 2007; Yang et al., 2012). L63 is often used to focus on the essence of the problem since L63 shows typical chaotic behaviors with the solution manifold being a well-known "butterfly attractor" (Fig. 1a), which has two regimes or wings corresponding to the positive and negative values for variable x. The regime shifts randomly, and the predictability is limited due to chaos. Evans et al. (2004) revealed predictability of the regime shift from rapidly growing uncertainties given by the growth rate of specific growing perturbations known as the bred vectors (Toth and Kalnay, 1993).

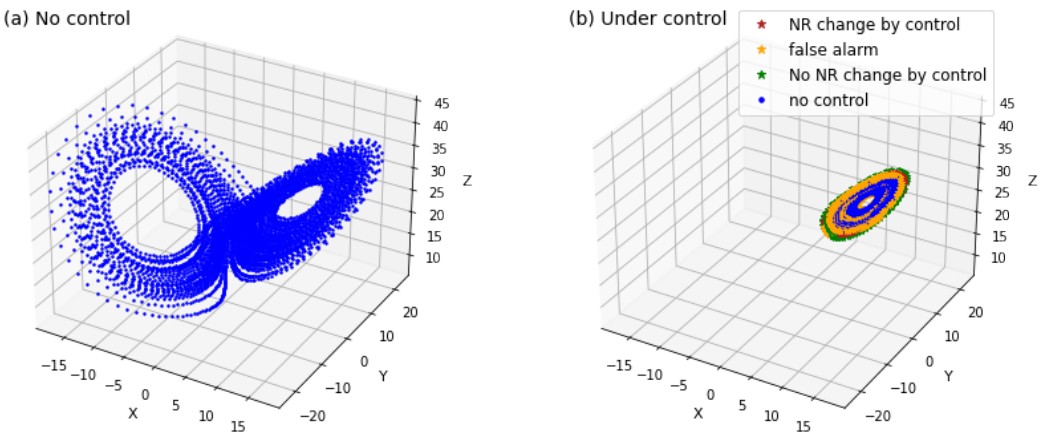

**Figure 1: Phase space of the 3-variable Lorenz model. (a) Lorenz's butterfly attractor from the NR without control, (b) the NR under control ($D=0.05$, $T=\lceil 4T_0 \rceil$). Each dot shows every time step for 8000 steps. See also a movie at https://doi.org/10.5446/54893.**

## 2 Experiments

We first perform a regular OSSE following the previous studies (Kalnay et al., 2007; Yang et al., 2012). The L63 system with the standard choice of the parameters (Lorenz, 1963) is discretized in time by the Runge-Kutta $4^{th}$ order scheme with a time step of 0.01 units. We define 1 step = 0.01 units throughout the paper. We assimilate observations every $T_a=8$ steps. A round of the orbit, i.e., from a maximum to the next maximum for variable x, corresponds to $T_0=75.1$ steps on average. We use the ensemble Kalman filter (EnKF, e.g., Evensen, 1994; Houtekamer and Zhang, 2016) with three ensemble members, which represent equally probable state estimates. For simplicity, we observe all three variables in this study, but any subset of observations except for observing only z variable results in the same conclusion, as suggested by the previous study about chaos synchronization (Yang et al., 2006). The observation noise is generated from the normal distribution for each variable

independently with the variance of 2.0. The EnKF results in accurate state estimation of the root mean square error (RMSE) of 0.32, consistently with the previous studies.

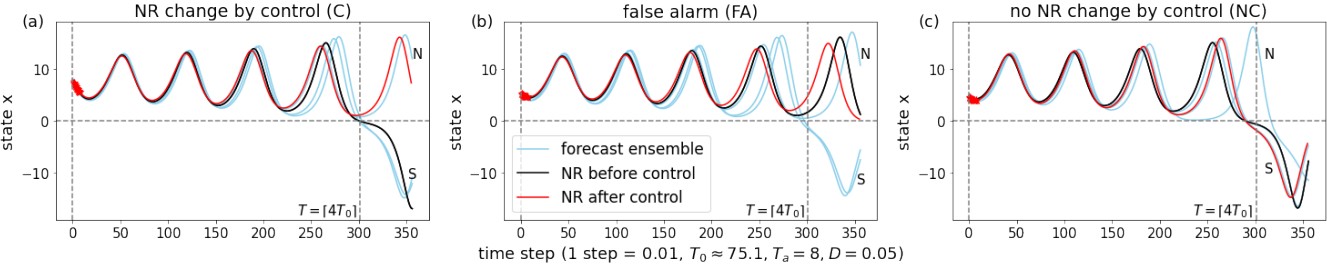

**Figure 2: Control cases with T=[4T₀] and D=0.05 for (a) NR changed (C), (b) false alarm (FA), (c) NR unchanged (NC). Red ticks at the beginning (t=1,…,7) shows adding perturbations to the NR.**

Next, we extend the OSSE and design a CSE. The goal of the control is to stay in a wing of the butterfly attractor without shifting to the other. It is essential that our prediction and control system is blind to the NR and takes only the imperfect observations. The control system finds when and what perturbations to add to the NR as follows (cf. Fig. 2):

1.    Perform a DA update using the observations at time t. (t=0 in Fig. 2)

2.    Run an ensemble forecast for T steps from time t to t+T. (T=$\lceil 4T_0 \rceil$ in Fig. 2, where $\lceil$ $\rceil$ indicates rounding up to the
closest integer since the model integration is discretized)

3.    If at least one ensemble member shows the regime shift, activate the control (step 4); otherwise, go to step 1 for the next DA at time t+$T_a$.

4.    Add perturbations with Euclidean norm D to the NR at every step from t+1 to t+$T_a$-1. More precisely, at time t+i (i=1,..,$T_a$-1), the NR state is evolved from the previous NR state at time t+i-1 and is perturbed by adding (dx, dy, dz)
where $\sqrt{dx^2 + dy^2 + dz^2} = D$. (Fig. 2 red ticks indicating perturbations added to the NR with D=0.05)

5.    At time t+$T_a$, the new NR is used to simulate the observations; go to step 1 for the next DA at time t+$T_a$.

Step 4 requires perturbations added to the NR. Investigating different strategies to generate the perturbations addresses controllability. Randomly chosen perturbations are found ineffective, but instead, we find the following strategy effective. We choose an ensemble member "S" showing the regime shift and another ensemble member "N" not showing the regime
shift. If all three ensemble members show the regime shift, we use the ensemble members from the former initial times for an extended forecasting period and identify an ensemble member "N" not showing the regime shift during the period from t to t+T. Take the differences of the two ensemble members S-N for every step from t+1 to t+$T_a$-1 (1 to 7 in Fig. 2) before the next observations are available at t+$T_a$ (8 in Fig. 2). The differences are used as perturbations added to the NR at appropriate time steps. Here, we consider the limitation of our intervention and include only a subset of the three variables (x, y, z) with
a limited perturbation size. The choice of the variables and norm D are the parameters for intervention.

Figure 2 illustrates three different cases with perturbations added to all three variables (x, y, z) with D=0.05 and T=$\lceil 4T_0 \rceil$. With these settings the control is successful as shown in Fig. 1 (b) for 8000 steps. Figure 2 (a) shows the case in which the

NR is changed by control and stays in the positive-x regime successfully (simply "C" for change). Figure 2 (b) shows the case of a false alarm (FA), in which the NR does not show the regime shift, but the ensemble prediction does. Therefore, the

perturbations are added unnecessarily, but do not hurt. Figure 2 (c) shows the case in which the NR is not changed by control and still shows the regime shift (simply "NC" for no change).

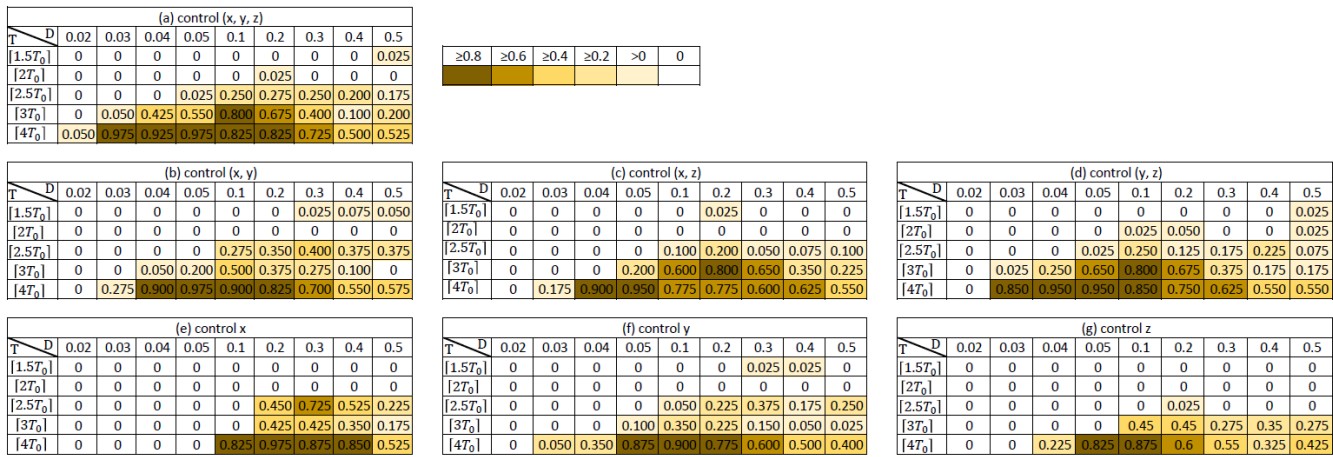

**Figure 3: Rates of successful control out of 40 CSEs for perturbations added to variables (a) x, y, z, (b) x, y, (c) x, z, (d) y, z, (e) x, (f) y, (g) z.**

To investigate the sensitivity to the parameters T and D, and the choice of the perturbed variables, we perform 40 independent experiments for each setting for 8000 steps (1000 DA cycles, cf. Appendix A for the exact choices of the initial conditions) and count the number of successful experiments in which the NR stays in a single regime under control. Higher success rates correspond to better controllability. With longer forecasts (larger T), control is generally more effective (Fig. 3). With small $T<[2.5T_0]$, the success rates are very low. The mean transition time for the regime shift is approximately $2.3T_0$,

which may be the minimum forecast length for effective control. With very small perturbations (D=0.02), the control is difficult, but larger D does not necessarily improve the success rate. The perturbations are added every step, and the state evolves by approximately 0.5 (Euclidean norm) in one step on average (Table 1). This is about a half of the evolution without control, suggesting that the perturbations effectively drag the NR states toward more stable regions of the attractor (cf. Fig. 1 and a movie at https://doi.org/10.5446/54893). Adding larger perturbations with a similar size to the one-step

model evolution tends to reduce the effect of control. Although observing only z is not sufficient for DA, it is good for control. Perturbing only one variable y or z is effective with $T=[4T_0]$ and D>0.04, only an eighth of the analysis error of 0.32 or only 3% of the observation error standard deviation of $\sqrt{2}$. In short, the L63 regime change is well controllable.

| D | 0.02 | 0.03 | 0.04 | 0.05 | 0.1 | 0.2 | 0.3 | 0.4 | 0.5 | no control |
|---|---|---|---|---|---|---|---|---|---|---|
| OME | 0.694 | 0.608 | 0.594 | 0.577 | 0.536 | 0.488 | 0.461 | 0.422 | 0.403 | 0.956 |
| D/OME | 0.029 | 0.049 | 0.067 | 0.087 | 0.186 | 0.410 | 0.651 | 0.947 | 1.239 | NA |

**Table 1: Averaged one-step model evolution in the Euclidean norm (OME) and the relative size of perturbations (D/OME). Only successful control cases are considered for CSEs with $T=[4T_0]$ and perturbations added to variables x, y, z.**

We further investigate the rates of false alarms (FA), NR changed (C) and unchanged (NC) by perturbations (Fig. 4a). With larger D, we find generally fewer interventions. With smaller D, we have more interventions mostly by FA. With smaller D, higher rates of NC suggest that longer-term small interventions be needed. Additional experiments by not applying FA and/or NC perturbations reveal relative importance of these perturbations (Fig. 4b). These experiments require knowing the NR T steps in advance and therefore are not practical but are useful to understand the roles of these

perturbations. For D=0.2 and smaller, not applying FA perturbations does not significantly contribute to the control (Fig. 4b, yellow), whereas not applying NC has significant impact on reducing the effect of control (Fig. 4b, blue, green). Namely, the accumulation of NC perturbations would be essential for effective control. With large D>0.2, not applying FA and NC perturbations significantly enhances the effect of control (Fig. 4b, green). With the perturbation size similar to or even larger than the one-step model evolution (Table 1), a single instance of C perturbations is quite significant. In these cases, FA and

NC perturbations are found harmful.

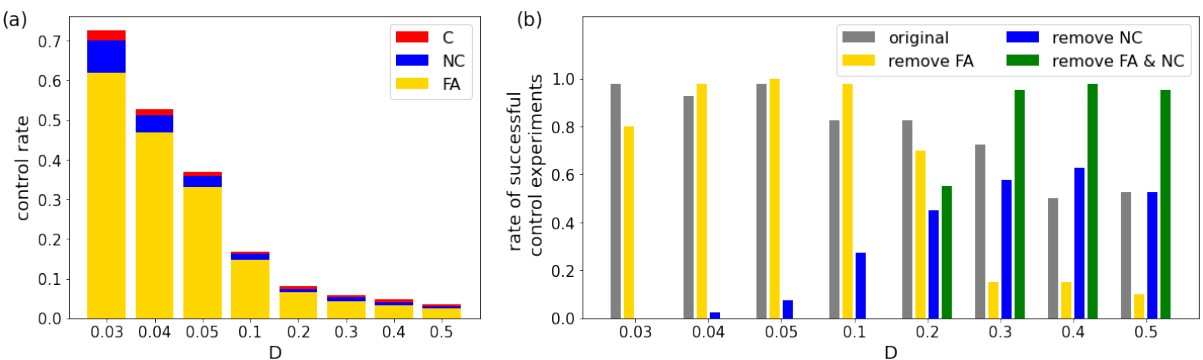

**Figure 4: (a) rates of the cases of NR changed (C, red), false alarm (FA, yellow), NR unchanged (NC, blue) for the successful control experiments with $T=[4T_0]$ and perturbations added to variables x, y, z. The rates indicate the number of cases out of total 1000 DA cycles. (b) rates of successful control experiments with $T=[4T_0]$ and perturbations added to variables x, y, z for the**
**original CSE (grey, cf. Fig. 3a), CSE without applying FA perturbations (yellow), CSE without applying NC perturbations (blue), CSE without applying FA and NC perturbations (green).**

      Finally, we perform additional sensitivity experiments with a longer DA interval of $T_a=25$ steps and with partial observations, i.e., only one or two variables are observed. The results generally agree with what has been shown so far (cf. Appendix B).

**3 Conclusions**

      In this study, we proposed the CSE with numerical demonstration using the L63 3-variable model. The OSSE is a well-known, powerful approach to study predictability and to evaluate DA methods and observing systems without having real-world observation data. The CSE is an extension to the OSSE to study controllability and can be applied to various dynamical systems including full-scale NWP models. Our future studies apply the CSE to more complex models and

investigate different control scenarios such as controlling the occurrences of extreme events. Such studies will address

critical issues like how manageable interventions in terms of cost and energy can make differences to extreme events. This study is only a small step toward broad investigations that may lead to effective control of weather events.

As we described in introduction, any real-world application requires extensive caution. For the case of the L63 model, one side of the attractor may not be desirable for all aspects. We must consider and assess every potential impact caused by
the control and have proper protocols for social, ethical, and legal agreement about real-world operations.

## Appendix A (The initial conditions of 40 CSEs)

The OSSE with the L63 model follows that of the previous studies (Kalnay et al., 2007; Yang et al., 2012; Miller et al., 1994; Evensen, 1997). Here we describe the additional details that were not provided in the previous papers but are necessary to repeat the experiment in this study. The initial condition for the NR was chosen to be $(x, y, z) = (8.20747939,$
$10.0860429, 23.86324441)$ after running the L63 model for 1000 steps initialized by the three state variables taken from independent random draws from a normal distribution with mean 0 and variance 2.0. The NR was 8 million steps long, and the OSSE was performed for the same period as the NR.

The CSEs were performed for total 378 combinations of T, D and the choice of the intervention. There were 9, 5, and 7 choices of T, D, and intervention as shown in Fig. 3. For each combination, 40 independent CSEs were performed for 8000
steps. The initial conditions for the 40 CSEs were chosen from the analyzed states of the OSSE at different time points as shown in Table 1. Figure 1 (b) shows the CSE #1 and Fig. 1 (a) the corresponding period of the NR.

| CSE index | Time point of the NR | CSE index | Time point of the NR | CSE index | Time point of the NR | CSE index | Time point of the NR |
|---|---|---|---|---|---|---|---|
| 1 | 106069 | 11 | 126902 | 21 | 150056 | 31 | 173894 |
| 2 | 107043 | 12 | 128058 | 22 | 150796 | 32 | 175011 |
| 3 | 109371 | 13 | 130718 | 23 | 152308 | 33 | 179671 |
| 4 | 111261 | 14 | 132342 | 24 | 155048 | 34 | 184480 |
| 5 | 112987 | 15 | 133311 | 25 | 155666 | 35 | 197270 |
| 6 | 114146 | 16 | 138699 | 26 | 162753 | 36 | 199278 |
| 7 | 122065 | 17 | 140562 | 27 | 164411 | 37 | 200712 |
| 8 | 124720 | 18 | 144953 | 28 | 168461 | 38 | 201304 |
| 9 | 125339 | 19 | 147614 | 29 | 172109 | 39 | 208511 |
| 10 | 125854 | 20 | 149418 | 30 | 173399 | 40 | 209397 |

Initial time point = Time point of the NR – T – [(Time point of the NR – T) mod $T_a$]

**Table A1: Time points of the NR providing the initial conditions of the 40 independent CSEs. The initial time point coincides the time with observations only every $T_a= 8$ steps, and the formula underneath the table provides the exact initial time point from the**
**value in the table for a given parameter of T.**

## Appendix B (Additional sensitivity experiments)

CSEs are performed with a longer DA interval of $T_a=25$ steps which result in the RMSE of 0.76, consistently with the previous studies. The results are generally consistent (Fig. B1 compared with Fig. 3).

| (a) control (x, y, z) | | | | | | | |
|---|---|---|---|---|---|---|---|
| T \ D | 0.03 | 0.04 | 0.05 | 0.1 | 0.2 | 0.3 | 0.4 | 0.5 |
| $[1.5T_0]$ | 0 | 0 | 0 | 0.025 | 0.05 | 0.025 | 0.025 | 0 |
| $[2T_0]$ | 0 | 0 | 0 | 0.725 | 0.575 | 0.275 | 0.025 | 0 |
| $[2.5T_0]$ | 0 | 0.175 | 0.75 | 0.975 | 0.675 | 0.85 | 0.425 | 0.05 |
| $[3T_0]$ | 0.275 | 0.975 | 1 | 1 | 1 | 0.725 | 0.425 | 0.1 |
| $[4T_0]$ | 1 | 1 | 1 | 1 | 1 | 0.825 | 0.375 | 0.025 |

Legend: ≥0.8  ≥0.6  ≥0.4  ≥0.2  >0  0

| (b) control (x, y) | | | | | | | |
|---|---|---|---|---|---|---|---|
| T \ D | 0.03 | 0.04 | 0.05 | 0.1 | 0.2 | 0.3 | 0.4 | 0.5 |
| $[1.5T_0]$ | 0 | 0 | 0 | 0 | 0.1 | 0.125 | 0.1 | 0.025 |
| $[2T_0]$ | 0 | 0 | 0 | 0.575 | 0.425 | 0.15 | 0.075 | 0 |
| $[2.5T_0]$ | 0 | 0 | 0 | 0.75 | 0.8 | 0.775 | 0.475 | 0.175 |
| $[3T_0]$ | 0 | 0.125 | 0.725 | 1 | 0.9 | 0.725 | 0.375 | 0.1 |
| $[4T_0]$ | 0.25 | 0.975 | 1 | 1 | 1 | 0.825 | 0.45 | 0.25 |

| (c) control (x, z) | | | | | | | |
|---|---|---|---|---|---|---|---|
| T \ D | 0.03 | 0.04 | 0.05 | 0.1 | 0.2 | 0.3 | 0.4 | 0.5 |
| $[1.5T_0]$ | 0 | 0 | 0 | 0 | 0.05 | 0.025 | 0.025 | 0 |
| $[2T_0]$ | 0 | 0 | 0 | 0.3 | 0.5 | 0.325 | 0.2 | 0.075 |
| $[2.5T_0]$ | 0 | 0 | 0 | 0.95 | 0.825 | 0.725 | 0.625 | 0.2 |
| $[3T_0]$ | 0 | 0 | 0.65 | 1 | 1 | 0.975 | 0.675 | 0.6 |
| $[4T_0]$ | 0.05 | 0.975 | 1 | 1 | 1 | 1 | 0.875 | 0.575 |

| (d) control (y, z) | | | | | | | |
|---|---|---|---|---|---|---|---|
| T \ D | 0.03 | 0.04 | 0.05 | 0.1 | 0.2 | 0.3 | 0.4 | 0.5 |
| $[1.5T_0]$ | 0 | 0 | 0 | 0 | 0.025 | 0.1 | 0.125 | 0.05 |
| $[2T_0]$ | 0 | 0 | 0 | 0.725 | 0.575 | 0.425 | 0.15 | 0.05 |
| $[2.5T_0]$ | 0 | 0 | 0.55 | 0.875 | 0.875 | 0.55 | 0.45 | 0.075 |
| $[3T_0]$ | 0.05 | 0.875 | 1 | 1 | 0.95 | 0.875 | 0.55 | 0.125 |
| $[4T_0]$ | 0.95 | 1 | 1 | 1 | 1 | 0.875 | 0.575 | 0.075 |

| (e) control x | | | | | | | |
|---|---|---|---|---|---|---|---|
| T \ D | 0.03 | 0.04 | 0.05 | 0.1 | 0.2 | 0.3 | 0.4 | 0.5 |
| $[1.5T_0]$ | 0 | 0 | 0 | 0 | 0 | 0.025 | 0.025 | 0 |
| $[2T_0]$ | 0 | 0 | 0 | 0 | 0.05 | 0.1 | 0.1 | 0 |
| $[2.5T_0]$ | 0 | 0 | 0 | 0 | 0.875 | 0.775 | 0.75 | 0.3 |
| $[3T_0]$ | 0 | 0 | 0 | 0.225 | 0.975 | 0.9 | 0.75 | 0.325 |
| $[4T_0]$ | 0 | 0 | 0 | 1 | 1 | 1 | 1 | 0.7 |

| (f) control y | | | | | | | |
|---|---|---|---|---|---|---|---|
| T \ D | 0.03 | 0.04 | 0.05 | 0.1 | 0.2 | 0.3 | 0.4 | 0.5 |
| $[1.5T_0]$ | 0 | 0 | 0 | 0 | 0.025 | 0.075 | 0.025 | 0.025 |
| $[2T_0]$ | 0 | 0 | 0 | 0.05 | 0.4 | 0.175 | 0.025 | 0 |
| $[2.5T_0]$ | 0 | 0 | 0 | 0.625 | 0.55 | 0.6 | 0.375 | 0.15 |
| $[3T_0]$ | 0 | 0 | 0.1 | 0.825 | 0.775 | 0.425 | 0.25 | 0.025 |
| $[4T_0]$ | 0 | 0.65 | 0.925 | 1 | 1 | 0.75 | 0.3 | 0.1 |

| (g) control z | | | | | | | |
|---|---|---|---|---|---|---|---|
| T \ D | 0.03 | 0.04 | 0.05 | 0.1 | 0.2 | 0.3 | 0.4 | 0.5 |
| $[1.5T_0]$ | 0 | 0 | 0 | 0 | 0 | 0 | 0 | 0 |
| $[2T_0]$ | 0 | 0 | 0 | 0 | 0.275 | 0.075 | 0.075 | 0 |
| $[2.5T_0]$ | 0 | 0 | 0 | 0.775 | 0.725 | 0.2 | 0 | 0 |
| $[3T_0]$ | 0 | 0 | 0 | 0.975 | 0.95 | 0.8 | 0.325 | 0.225 |
| $[4T_0]$ | 0 | 0.375 | 1 | 1 | 0.975 | 0.775 | 0.35 | 0.4 |

Figure B1: Similar to Fig. 3, but for the case with a longer DA interval of $T_a$=25.

CSEs are performed with different observing coverages, and the results are summarized in Table B1. Multiplicative inflation is manually tuned for each observing coverage.

| D | obs x | obs y | obs x, y | obs x, z | obs y, z | obs x, y, z |
|---|---|---|---|---|---|---|
| 0.02 | 0 | 0.025 | 0.05 | 0.125 | 0 | 0.05 |
| 0.03 | 1 | 0.95 | 0.95 | 0.975 | 0.975 | 0.975 |
| 0.04 | 1 | 0.975 | 0.95 | 1 | 1 | 0.925 |
| 0.05 | 1 | 1 | 0.975 | 1 | 1 | 0.975 |
| 0.1 | 1 | 1 | 1 | 1 | 1 | 0.825 |
| 0.2 | 0.975 | 0.925 | 0.85 | 0.975 | 0.975 | 0.825 |
| 0.3 | 0.95 | 0.925 | 0.675 | 0.975 | 0.95 | 0.725 |
| 0.4 | 0.95 | 0.8 | 0.78 | 0.975 | 0.875 | 0.5 |
| 0.5 | 0.9 | 0.75 | 0.65 | 0.95 | 0.85 | 0.525 |
| Ensemble spread | 0.807 | 0.469 | 0.376 | 0.477 | 0.323 | 0.27 |
| RMSE | 0.908 | 0.507 | 0.412 | 0.564 | 0.356 | 0.32 |
| multiplicative inflation | 1.065 | 1.05 | 1.045 | 1.09 | 1.06 | 1.04 |

Table B1: Rates of successful control out of 40 CSEs with different observing coverage. T=$[4T_0]$ and perturbations are added to variables x, y, z.

## Code availability

The code that supports the findings of this study are available from the corresponding author upon reasonable request.

**Data availability**

The authors declare that all data supporting the findings of this study are available within the paper.

**Author contribution**

TM is the principal investigator, directed the research and prepared the manuscript with contributions from QS. QS performed numerical experiments and visualized the results.

**Competing interests**

Takemasa Miyoshi is an editor of NPG.

**Acknowledgements**

This study was partly supported by the RIKEN Junior Research Associate (JRA) program and by the Japan Science and Technology Agency (JST) Moonshot R&D Millenia program (grant number JPMJMS20MK).

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
