# Peer review of "Control Simulation Experiment with the Lorenz's Butterfly Attractor"

_Nonlinear Processes in Geophysics, 2021_

## Author Comment (AC1)

**RC1**
We thank the reviewer for her/his kind review and constructive comments.

General comment
This manuscript is devising an approach to control the trajectory of the Lorenz system in one specific wing of its attractor. The approach is based on the perturbing the solutions once some forecasts show transitions toward the other wing. The idea behind their approach is to control the weather in order to reduce disaster risks.

The approach proposed is interesting but I think that the authors are presenting their method as a positive way to reduce risks although it could be completely disastrous. A perturbation at some location in the world can suppress or generate a thunderstorm elsewhere. In the idealized setting of this manuscript, from the point of view of a certain observer, to be in one wing of the attractor is disastrous (e.g. not enough rain), but for another observer this could be the opposite (more rain expected). So selling that the method will solve the problem of risks is to my opinion not appropriate. The authors should be more balanced on this, and add cautious comments in the introduction and in the conclusions.
RESPONSE: We perfectly agree and will address this point with caution in the revised manuscript.

Another major point not discussed in the manuscript is how this type of perturbations could be made in practice. Imagine that one wants to reduce the risk of a hurricane or a Typhoon flooding a region. We can imagine to reduce the temperature of the ocean. What energy do we need to do that and how to do that? My guess is that it is probably a huge amount of energy. The author should discuss the applicability of their method in the conclusions in the perspective of their results.

Again in the previous example of the Hurricane or Typhoon, the impact of such perturbation could lead to considerable problems elsewhere, and the benefit for it could not be oversold and even misleading.
RESPONSE: We perfectly agree and will add discussions about this point in the revised manuscript.

Specific comments:

- At page 4, lines 98-105. The authors present the way they perturb the system. In step 4 they are indicating the way the perturbation is introduced. Please provide equations for that to clarify where the parameter D and the perturbations are appearing.
RESPONSE: We will revise accordingly.

- From a dynamical perspective, I am wondering what is the nature of the successful perturbations (aligned along the stable, unstable or neutral directions…). Please comment on that.

RESPONSE: The successful control case (Fig. 1b) shows that the NR under control is confined in a more stable region than the case without control (Fig. 1a). The results suggest that the successful perturbations draw the NR states toward more stable regions of the attractor. We will revise the manuscript accordingly with an additional supplementary material of a movie showing the side-by-side comparison of the NRs with/without control in the three-dimensional phase space.

- Certain periods, T, seem to be more favorable for the control. Is there any relation with the characteristic time scales of the dynamics, the mean transition time between the two wings for instance?   This should be discussed.

RESPONSE: Thank you very much for the very insightful suggestion, which led to an interesting finding as follows. We computed the mean transition time for the regime shift and found it to be approximately 2.3 $T_0$. Figure 3 indicates clearly T = 2.5 $T_0$ or larger is a favorable choice for successful control. We will add this discussion in the revised manuscript.

- What is the impact of changing the DA cycle and the absence of observations of some variables?

RESPONSE: We performed additional experiments with a longer observation interval of 25 steps and with partial observations. The results showed generally consistent conclusion. We will add these results in the revised manuscript.

**RC2**

We thank the reviewer for her/his kind review and constructive comments.

General comments

The manuscript "Control Simulation Experiment with the Lorenz's Butterfly Attractor" by T. Miyoshi and Q. Sun proposes very interesting experiments that possibly give some hints to control chaotic dynamical systems of nature in the future, although this reviewer is not sure of any circumstance, when everyone can agree with one direction of weather/climate control in the real world with complex interests.

RESPONSE: We perfectly agree and will address this point with caution in the revised manuscript.

Despite, scientific approach on this topic is fascinating indeed. Starting with OSSE, authors tried to find perturbations that suppress regime shift of the nature run and saw how well the nature run can be controlled by adding perturbations with different parameters.

The manuscript describes methodology and result too concisely, in general. There are many parts that need more explanations and clarifications. First, show how and where the norm D is applied explicitly.

RESPONSE: We will clarify this point in the revised manuscript.

Second, do the successful control cases of Figure 3 include the cases of Figure 2 (a) and (b) (the cases where nature run did not change the regime due to adding perturbations and the other cases of so-called false alarm)? If yes, it is confusing because the results of Figure 4 do not look consistent with those of Figure 3. Small ratio of successful control experiments should have dark green dots, but it does not look like that. Figure 4 does not deliver the findings effectively. If there is any misunderstanding of this reviewer, please fix it and clarify your description. Please give more descriptive analysis of results for readers to understand well.

RESPONSE: We apologize for the lack of descriptions about what are exactly shown in the figures. We will revise the manuscript accordingly.

Third, Figure 4 shows incredibly small ratio of NR changed cases (magenta). Does it mean that adding perturbation is very unlikely to control the NR indeed, doesn't it? Then, the authors attempted those interesting experiments but it was not easy to find perturbations to control the nature. Is this what we can conclude?

RESPONSE: Thank you very much for the very insightful comment, which led to additional investigations about the role of perturbations which do not lead to NR change (NC) and of false alarm (FA) perturbations. The perturbations are designed to draw the NR state to the

stable regions of the attractor and are effective for that purpose. To identify the relative roles of the NC and FA perturbations, we performed additional experiments by removing these perturbations. The results showed that the NC perturbations play an essential role in successful control, and that it is more so for small perturbation amplitudes (i.e., small D). Namely, the NR is changed to the desired direction after the accumulation of perturbations that do not directly lead to the NR change. We will add the additional results and discussions in the revised manuscript.

Besides, please rephrase the definition of the ratio in Figure 4, which is not easy to intuitively understand. Maybe inconsistency between Figure 3 and 4 came from lack of explanation. Please discuss more about what you have obtained from those experiments. This reviewer is not sure whether this figure can deliver the result well. Please consider a table with numbers or other type of graph. RESPONSE: We apologize again for the lack of descriptions about what are exactly shown in the figures. We will revise the manuscript accordingly.

This reviewer understands that the manuscript provides an interesting and novel idea of CSE helping one study how to control the chaos. Still, experimental results are not analyzed enough and hence the methodology shown here does not look very effective. This is this reviewer's opinion.

Hope this reviewer's comments help this manuscript improved well enough for the publication. RESPONSE: Thank you very much for the valuable, constructive comments. We will address the comments and improve the manuscript.

**CC1:**

We thank Paul PUKITRE for his kind review and constructive comments.

The topic of the paper seems to be the role of a forced response. From Osipov, G. V., Kurths, J., & Zhou, C. (2007). Synchronization in oscillatory networks. Berlin: Springer, it is known that a periodic forcing can reduce the erratic fluctuations and uncertainty of a near‑chaotic response function. This is an important path of research when one considers that natural climate is continually corrected by periodic forcing such as the daily signal, seasonal/annual signal, and the more complex tidal forcing. So that even with a weakly non-linear formulation, the response may not easily be decomposed but it also may not be chaotic. Thus, there may be hope in deconstructing seemingly erratic time-series such as ENSO.

RESPONSE: We studied the book and would like to include the relevant discussion in the revised manuscript.

---

## Author Response (AR1)

**RC1**

We thank the reviewer for her/his kind review and constructive comments.

General comment

This manuscript is devising an approach to control the trajectory of the Lorenz system in one specific wing of its attractor. The approach is based on the perturbing the solutions once some forecasts show transitions toward the other wing. The idea behind their approach is to control the weather in order to reduce disaster risks.

The approach proposed is interesting but I think that the authors are presenting their method as a positive way to reduce risks although it could be completely disastrous. A perturbation at some location in the world can suppress or generate a thunderstorm elsewhere. In the idealized setting of this manuscript, from the point of view of a certain observer, to be in one wing of the attractor is disastrous (e.g. not enough rain), but for another observer this could be the opposite (more rain expected). So selling that the method will solve the problem of risks is to my opinion not appropriate. The authors should be more balanced on this, and add cautious comments in the introduction and in the conclusions.

RESPONSE: We perfectly agree and added descriptions in the revised manuscript (L.54-55, L.168-170).

Another major point not discussed in the manuscript is how this type of perturbations could be made in practice. Imagine that one wants to reduce the risk of a hurricane or a Typhoon flooding a region. We can imagine to reduce the temperature of the ocean. What energy do we need to do that and how to do that? My guess is that it is probably a huge amount of energy. The author should discuss the applicability of their method in the conclusions in the perspective of their results.

Again in the previous example of the Hurricane or Typhoon, the impact of such perturbation could lead to considerable problems elsewhere, and the benefit for it could not be oversold and even misleading.

RESPONSE: We perfectly agree and added discussion about this point in the revised manuscript (L.165-166).

Specific comments:

- At page 4, lines 98-105. The authors present the way they perturb the system. In step 4 they are indicating the way the perturbation is introduced. Please provide equations for that to clarify where the parameter D and the perturbations are appearing.

RESPONSE: Revised accordingly (L.103-105).

- From a dynamical perspective, I am wondering what is the nature of the successful perturbations (aligned along the stable, unstable or neutral directions…). Please comment on that.

RESPONSE: The successful control case (Fig. 1b) shows that the NR under control is confined in a more stable region than the case without control (Fig. 1a). The results suggest that the successful perturbations drag the NR states toward more stable regions of the attractor. We added the description in the revised manuscript (L.132-134, Table 1). In addition, to show more clearly how the NR stays in more stable regions, we added a DOI link to the movie showing the side-by-side comparison of the NRs with/without control in the three-dimensional phase space.

- Certain periods, T, seem to be more favorable for the control. Is there any relation with the characteristic time scales of the dynamics, the mean transition time between the two wings for instance?   This should be discussed.

RESPONSE: Thank you very much for the very insightful suggestion, which led to an interesting finding as follows. We computed the mean transition time for the regime shift and found it to be approximately $2.3 T_0$. Figure 3 indicates clearly $T = 2.5 T_0$ or larger is a favorable choice for successful control. We added the description in the revised manuscript (L.129-130).

- What is the impact of changing the DA cycle and the absence of observations of some variables?

RESPONSE: We performed additional experiments with a longer observation interval of 25 steps and with partial observations. The results showed generally consistent conclusion. We added the descriptions (L.157-159) and the results in Appendix B in the revised manuscript.

We thank the reviewer for her/his kind review and constructive comments.

General comments

The manuscript "Control Simulation Experiment with the Lorenz's Butterfly Attractor" by T. Miyoshi and Q. Sun proposes very interesting experiments that possibly give some hints to control chaotic dynamical systems of nature in the future, although this reviewer is not sure of any circumstance, when everyone can agree with one direction of weather/climate control in the real world with complex interests.

RESPONSE: We perfectly agree and added descriptions in the revised manuscript (L.54-55, L.168-170).

Despite, scientific approach on this topic is fascinating indeed. Starting with OSSE, authors tried to find perturbations that suppress regime shift of the nature run and saw how well the nature run can be controlled by adding perturbations with different parameters.

The manuscript describes methodology and result too concisely, in general. There are many parts that need more explanations and clarifications. First, show how and where the norm D is applied explicitly.

RESPONSE: We added more precise description in the revised manuscript (L.103-105).

Second, do the successful control cases of Figure 3 include the cases of Figure 2 (a) and (b) (the cases where nature run did not change the regime due to adding perturbations and the other cases of so-called false alarm)? If yes, it is confusing because the results of Figure 4 do not look consistent with those of Figure 3. Small ratio of successful control experiments should have dark green dots, but it does not look like that. Figure 4 does not deliver the findings effectively. If there is any misunderstanding of this reviewer, please fix it and clarify your description. Please give more descriptive analysis of results for readers to understand well.

RESPONSE: We apologize for the lack of descriptions about what are exactly shown in the figures. We replaced Fig. 4 and added related descriptions (L.140-150) for more clarity and more findings in the revised manuscript.

Third, Figure 4 shows incredibly small ratio of NR changed cases (magenta). Does it mean that adding perturbation is very unlikely to control the NR indeed, doesn't it? Then, the authors attempted those interesting experiments but it was not easy to find perturbations to control the nature. Is this what we can conclude?

RESPONSE: Thank you very much for the very insightful comment, which led to additional investigations about the role of perturbations which do not lead to NR change (NC) and of

false alarm (FA) perturbations. The perturbations are designed to drag the NR state to the stable regions of the attractor and are effective for that purpose. To identify the relative roles of the NC and FA perturbations, we performed additional experiments by removing these perturbations. The results showed that the NC perturbations play an essential role in successful control, and that it is more so for small perturbation amplitudes (i.e., small D). Namely, the NR is changed to the desired direction after the accumulation of perturbations that do not directly lead to the NR change. We added the additional results and discussions in the revised manuscript (Fig. 4b, L.140-150).

Besides, please rephrase the definition of the ratio in Figure 4, which is not easy to intuitively understand. Maybe inconsistency between Figure 3 and 4 came from lack of explanation. Please discuss more about what you have obtained from those experiments. This reviewer is not sure whether this figure can deliver the result well. Please consider a table with numbers or other type of graph.
RESPONSE: We apologize again for the lack of descriptions about what are exactly shown in the figures. We revised the manuscript accordingly (Fig. 3, Fig. 4, L140-150).

This reviewer understands that the manuscript provides an interesting and novel idea of CSE helping one study how to control the chaos. Still, experimental results are not analyzed enough and hence the methodology shown here does not look very effective. This is this reviewer's opinion.

Hope this reviewer's comments help this manuscript improved well enough for the publication.
RESPONSE: Thank you very much for the valuable, constructive comments which helped significantly improve the manuscript.